# Berberine potentiates liver inflammation and fibrosis in the PI*Z hAAT transgenic murine model

Yuanqing Lu, Naweed S. Mohammad, Jungnam Lee, Alek M. Aranyos, Karina A. Serban, Mark L. Brantly *

Division of Pulmonary, Critical Care and Sleep Medicine, Department of Medicine, University of Florida, Gainesville, Florida, United States of America

* mbrantly@ufl.edu

## Abstract

### Background

Alpha-1 antitrypsin deficiency (AATD) is an inherited disease, the common variant caused by a Pi*Z mutation in the SERPINA1 gene. Pi*Z AAT increases the risk of pulmonary emphysema and liver disease. Berberine (BBR) is a nature dietary supplement and herbal remedy. Emerging evidence revealed that BBR has remarkable liver-protective properties against various liver diseases. In the present study, we investigated the therapeutic effects and toxicities of BBR in Pi*Z hepatocytes and Pi*Z transgenic mice.

### Methods

Huh7.5 and Huh7.5Z (which carries the Pi*Z mutation) cells were treated with different concentrations of BBR for 48 hours. MTT was performed for cell viability assay. Intracellular AAT levels were evaluated by western blot. In vivo studies were carried out in wild type, native phenotype AAT (Pi*M), and Pi*Z AAT transgenic mice. Mice were treated with 50 mg/kg/day of BBR or solvent only by oral administration for 30 days. Western blot and liver histopathological examinations were performed to evaluate therapeutic benefits and liver toxicity of BBR.

### Results

BBR reduced intracellular AAT levels in Huh7.5Z cells, meanwhile, no Pi*Z-specific toxicity was observed. However, BBR did not reduce liver AAT load but significantly potentiated liver inflammation and fibrosis accompanying the activation of unfolded protein response and mTOR in Pi*Z mice, but not in wild type and Pi*M mice.

### Conclusions

BBR exacerbated liver inflammation and fibrosis specifically in Pi*Z mice. This adverse effect may be associated with the activation of unfolded protein response and mTOR. This study implicates that BBR should be avoided by AATD patients.

**Data Availability Statement:** All relevant data are within the manuscript and its Supporting information files.

**Funding:** MB. The Alpha-1 Foundation Research Professorship #007319. The Alpha-1 Foundation had no role in study design, data collection and analysis, decision to publish, or preparation of the manuscript.

**Competing interests:** The authors have declared that no competing interests exist.

## Introduction

Berberine (BBR) is an isoquinoline alkaloid that is found in many plants. It has been used in traditional Chinese medicine and Ayurvedic medicine for thousands of years for its anti-diarrheal and anti-inflammatory properties [1]. Today, BBR is one of the commonly used dietary supplements and alternative medicine [2]. BBR is well-studied in preclinical and clinical studies, where it has displayed multiple pharmacological properties including anti-hyperglycemic and inflammatory [3], anti-cancer [4], anti-arrhythmic [5], anti-hypertensive [6], anti-oxidative [7], hypolipidemic [8], and many other therapeutic potentials [9].

Alpha-1 antitrypsin (AAT) is a serine protease inhibitor (PI). It is encoded by the SERPINA1 gene and primarily produced in the liver. Alpha-1 antitrypsin deficiency (AATD) is an inherited disease caused by SERPINA1 gene mutations. Among the mutations, a point mutation at codon 342 (Glu342Lys) in exon 5, known as Pi*Z mutation, is the primary disease-causing mutation, which is associated with the most common and severe AATD [10]. Compared with the native allele (Pi*M), Pi*Z AAT tends to be polymerized and accumulated in endoplasmic reticulum (ER) of hepatocytes, leading to ER stress and liver disease. About 10% to 35% of Pi*Z AATD patients develop liver damages, including liver inflammation, fibrosis, cirrhosis, and hepatocellular carcinoma [11–15]. There is no specific treatment for AATD-associated liver disease except liver transplantation for end-stage liver disease. Multiple attempts to combat AATD-associated liver disease have been tested, including siRNA-based therapy [16], small molecular protein chaperones [17], and augmentation of autophagy [18], but none of them have been approved for AATD treatment to date.

The unfolded protein response (UPR) instigates a transcriptional and translational response to ER stress, and it is composed of three branches: protein kinase R-like endoplasmic reticulum kinase (PERK), inositol-requiring enzyme 1α (IRE1α), and activating transcription factor 6α (ATF6α). UPR is initiated as a pro-survival response to enhance protein folding capacity and autophagic degradation, therefore, to reduce the accumulation of unfolded or misfolded proteins and restore ER function. Severe, prolonged and irreversible ER stress switches UPR to a pro-apoptotic signaling event that leads to cell death and inflammation [19]. If this pro-apoptotic response does not kill the cells and the ER stress is persistent, UPR, especially the pro-apoptosis branches like PERK and IRE1α, will be attenuated [20–22]. This adaptation is vulnerable and easily compromised by the factors that can disturb the ER homeostasis, or so-called second hit [22, 23]. UPR is the master regulator to reduce protein accumulation in the ER and maintain ER homeostasis. Recently, our study revealed the activation and adaptive suppression of UPR to PI*Z AAT in human hepatocellular and murine models [22], suggesting that UPR may be serve as a therapeutic target for treatment of AATD-associated liver disease.

It has been reported that BBR reduces ER stress and UPR [24]. It also enhances autophagy [25], and exhibits protective effects against liver fibrosis [26] and fatty liver [27]. We hypothesize that BBR may have the potential to reduce intracellular AAT burden in hepatocytes, therefore, to ameliorate AATD-associated liver disease. On the other hand, to reduce the risk and prevent AATD-associated liver disease, it is more important to identify and avoid the hepatotoxic substances for AATD patients. BBR is mainly distributed and metabolized in the liver [28]. As a common dietary supplement, BBR is widely available for AATD patients, but its safety and liver toxicity for AATD patient is unknown.

In this study, we investigated the therapeutic effect and liver toxicity of BBR in a novel Pi*Z AAT hepatocellular model and a Pi*Z AAT transgenic murine model.

## Material and methods

### Cell culture and treatment

Recently, by using CRISPR/Cas9 technique, we generated a hepatocyte cell line harboring a Pi*Z mutation from Huh7.5 cells, named Huh7.5Z [22]. Huh7.5Z and Huh7.5 cells were cultured on 12-well plates with high glucose (4.5 g/L) Dulbecco's Modified Eagle Medium (DMEM) (Cat# 11995, Gibco, Grand Island, NY, USA) supplementing with 10% FBS (Cat# 12306C, MilliporeSigma, Burlington, MA, USA), at 37°C in a humidified incubator with 5% $CO_2$. Cells were treated with different concentrations of berberine chloride (Cat# PHR1502, MilliporeSigma, Burlington, MA, USA) in Dimethyl sulfoxide (DMSO) (Cat# 1211006USP, MilliporeSigma, Burlington, MA, USA) or the same volume of DMSO as negative control (n = 3) once cell confluency reached 70%. The DMSO volume was ≤ 0.1% of the volume of cell culture medium. After 48 hours of treatment, cells were lysed for protein and RNA extractions. The conditioned media were collected for ELISA assay.

### Cell viability assay

Huh7.5Z and Huh7.5 cells in a 12-well plate were treated with berberine chloride as described above (n = 3). After 48 hours of treatment, 1 ml/well of 0.5 mg/ml of Thiazolyl blue tetrazolium bromide (MTT) (Cat# T-030-1, Golden Biotech, Jersey City, NJ, USA) in RPMI without phenol red was added into cell culture wells, and after 3 hours of incubation, the MTT solution was replaced with 150 μL of DMSO in each well to dissolve the cells. The absorbance at 570 nm was measured using a SpectraMax® M3 Multi-Mode Microplate Reader. The results were calculated as a percentage in relation to the mean OD of untreated cells.

### Animal experiment

Previously, we created two human AAT (hAAT) transgenic mouse lines on C57BL6/J background, and they are now available from The Jackson Laboratory (Bar Harbor, ME, USA). PI*M transgenic mice, formal name C57BL/6J-Tg (SERPINA1)1Mlb/J, Jackson lab stock No.037669, carry the native human SERPINA1 gene. Pi*Z transgenic mice, formal name C57BL/6J-Tg (SERPINA1*E366K) 1Mib/J, Jackson lab stock No. 037670, carry the SERPINA1 gene with the Pi*Z mutation. Transgenic mice and their wild-type (WT) littermates were bred under specific pathogen-free conditions at the University of Florida animal care and services facility at 22 ± 3°C with a 12:12 hour light-dark cycle. Mice were housed in ventilated cages with an automatic watering system with a standard diet (Envigo 2918), corn cob bedding, and standard enrichment. Animal studies were approved by the University of Florida Institutional Animal Care and Use Committee.

Mice were treated by oral gavage since BBR is often administered orally by human and laboratory animal, although the oral bioavailability of BBR is low [28]. We chose a medium dose of BBR according to the literature [8, 29–31]. To investigate the liver toxicity and the potential therapeutic effect of BBR, 6-to 7-month-old Pi*Z transgenic mice were randomly divided into two groups: control group (n = 7, 4 females and 3 males) and BBR treatment group (n = 7, 5 females, 2 males). The average mouse bodyweight was 27.9 ± 6.3 g. BBR treatment mice were treated by oral gavage with 50 mg/kg/day of berberine chloride (Cat# PHR1502, MilliporeSigma, Burlington, MA, USA) in 10% DMSO (Cat# 1211006USP, MilliporeSigma, Burlington, MA, USA) and 90% (20% SBE-β-CD in saline), while control mice were treated by oral gavage with 10 ml/kg/day of 10% DMSO + 90% (20% SBE-β-CD in saline). Sulfobutylether-β-Cyclodextrin (SBE-β-CD) was purchased from MedChemExpress (Monmouth Junction, NJ, USA), Cat# HY-17031. Mice were treated once a day for 30 days. At the end of the experiment, mice

were euthanized via carbon dioxide inhalation, followed by cervical dislocation for tissue collection. A repetitive animal experiment was conducted to confirm and extend the results from the first animal experiment. In this repetitive experiment, 4- to 6-month-old WT, Pi*M and Pi*Z mice of both genders were randomly divided to two groups: control and BBR treatment group. The average mouse bodyweight was 27.8 ± 4.0g. The animal sample size was as follows: WT control, n = 3; WT treatment, n = 3; Pi*M control, n = 5, Pi*M treatment, n = 6; Pi*Z control, n = 5; and Pi*Z treatment, n = 9. Mice received the same treatment as mentioned above. At the end of the experiments, mice were euthanized as mentioned above, liver tissues were harvested and stabilized in Allprotect Tissue Reagent (Cat# 76405, Qiagen, Germantown, MD, USA) and then stored at −80˚C for protein extraction. For histopathological examinations, liver tissues were fixed with 10% formalin for 12 hours.

## Enzyme-linked immunosorbent assay (ELISA)

The hAAT concentration in cell culture medium was measured by ELISA as previously described [32]. Briefly, Clear Flat-Bottom Immuno Nonsterile 96-Well Plates (Cat# 3855, ThermoFisher Scientific, Waltham, MA, USA) were coated with goat anti-hAAT antibody (Cat# 55111, MP Pharmaceuticals, Santa Ana, CA, USA). Purified hAAT (NIBSC, Ridge, Herts, UK) was used as the standard. Polyclonal rabbit anti-hAAT (Cat# A0012, Dako, Bath, UK) and goat anti-rabbit IgG conjugated with peroxidase (Cat# 1706515, Bio-Rad, Hercules, CA, USA) were used as primary and secondary antibodies. An O-phenylenediamine tablet (Cat# p9187, Sigma-Aldrich, St. Louis, MO, USA) was used as chromogenic substrate. ELISA plates were read on a SpectraMax M3 Multi-Mode Microplate Reader. Results were expressed as mean ± SD, n = 3.

## Quantitative real-time PCR (RT-qPCR)

SERPINA1 RNA levels in Huh7.5Z cells were measured by qPCR. Total RNA was purified with the RNeasy Plus Mini Kit (Cat# 74134, Qiagen, Germantown, MD, USA). The High-Capacity RNA-to-cDNA Kit (Cat# 4387406, ThermoFisher Scientific, Waltham, MA, USA) was used for the reverse transcription PCR. Gene expression levels were measured by RT-qPCR using an Applied Biosystems 7500 Fast Real-Time PCR System (Life Technologies, Carlsbad, CA, USA). The TaqMan™ Fast Advanced Master Mix (Cat# 44-445-57, Fisher scientific, Hampton, NH, USA) was used with the Applied Biosystems SERPINA probe (ID Hs00165475_m1, Cat# 169784, ThermoFisher Scientific, Waltham, MA, USA) and the eukaryotic 18S rRNA probe (VIC™/TAMRA™) (Cat# 4310893E, ThermoFisher Scientific, Waltham, MA, USA) as SERPINA1 probe and endogenous control probe. The results were presented as relative quantification as determined by the $2^{-\Delta\Delta Ct}$ method.

## Western blot analysis

The levels of target proteins in Huh7.5Z cells and mouse liver tissues were measured by western blot. Cellular and liver tissue lysates were prepared in radioimmunoprecipitation assay (RIPA) buffer (Cat# 89901, ThermoFisher Scientific, Waltham, MA, USA) containing protease/phosphatase inhibitor (Cat# 5872S Cell Signaling Technology, Danvers, MA, USA) and Ambion™ DNase I (RNase-free) (Cat# AM2222, ThermoFisher Scientific, Waltham, MA, USA). Tissue samples were subsequently homogenized using a 2010 Geno/Grinder homogenizer (SPEX Sample Prep, Metuchen, NJ, USA). Both cell and tissue samples were sonicated in a bath sonicator (Avanti polar lipids, Alabaster, AL, USA) for 2 minutes (for cell lysates) or 5 minutes (for tissue lysates). Samples were mixed with 2x Laemmli sample loading buffer (Cat# 1610737, Bio-Rad, Hercules, CA, USA) containing 5% 2-mercaptoethanol (Cat# 1610710, Bio-

Rad, Hercules, CA, USA) in 1:1 ratio and then heat-denatured at 95˚C for 10 minutes. Criterion TGX 4% to 15% Precast Midi protein gels (Cat# 4561086 Bio-Rad, Hercules, CA, USA) were used for electrophoresis and 0.45 μm or 0.2 μm (for low molecular weight proteins) nitrocellulose membranes (Cat# 1620115 and 1620112, Bio-Rad, Hercules, CA, USA) were used for transferring. Membranes were blocked with 5% nonfat dry milk (Cat# 1706404, Bio-Rad, Hercules, CA, USA) or 5% fetal bovine serum (BSA) (Cat# A3803, MilliporeSigma, Burlington, MA, USA) according to manufacturer's instruction and incubated with the primary antibody (Table in S1 Table) in blocking buffer at 1:1000 (1:5000 for GAPDH and hAAT antibodies) dilution. The membranes were washed and incubated with the goat anti-rabbit IgG antibody conjugated with horseradish peroxidase (Cat# 1706515, Bio-Rad, Hercules, CA, USA. 1:10000 dilution). Chemiluminescent developments were performed either by Clarity Western ECL Substrate (Cat# 1705062, Bio-Rad, Hercules, CA, USA) or Lumigen ECL Ultra TMA-6 (Cat# TMA-6, Lumigen, Southfield, MI, USA) for detection of weak bands. Membranes were imaged with a ChemiDoc Touch imaging system (Bio-Rad, Hercules, CA, USA). Relative quantification was performed using Image Lab (Bio-Rad, Hercules, CA, USA) and ImageJ software (NIH, Bethesda, MD, USA).

### Immunofluorescence (IF) staining

The intracellular AAT polymer was detected by IF staining. Huh7.5Z cells were cultured on a Millicell® EZ SLIDE 8-well glass slide (Cat# PEZGS0816, MilliporeSigma, Burlington, MA, USA) and treated with berberine chloride or DMSO as described above (n = 3). Cells were fixed with 4% formaldehyde for 10 minutes and then were permeabilized with 0.1% of Triton X-100 for 7 minutes. After permeabilization, cells were blocked with 2% BSA (Cat# A3803, MilliporeSigma, Burlington, MA, USA) for 45 minutes and then incubated with the mouse monoclonal antibody clone 2C1 (Cat# HM2289, Hycult Biotech, Wayne, PA, USA, 1:200 dilution), which recognizes the polymeric form of human alpha-1 antitrypsin, for 1 hour at room temperature. The 2C1 antibody was omitted in negative control wells. Incubated cells with goat anti-Rabbit IgG H&L Alexa Fluor 647 (Cat# ab150113, Abcam, Cambridge, UK, 1:800 dilution) at room temperature for 1 hour. After washing and airdrying, a coverslip was mounted with a drop of mounting medium with DAPI (Cat# DUO82040, MilliporeSigma, Burlington, MA, USA). Images were observed and scanned by a BZ-X710 all-in-one fluorescence microscope and quantification analysis was performed by BZ-X analysis software (Keyence, Osaka, Japan).

### Histological examination

The hematoxylin and eosin (H&E), CD3, myeloperoxidase (MPO), F4/80, Picrosirius Red, periodic acid–Schiff–diastase (PAS-D) and 2C1 stains on mouse liver slides were conducted by the University of Florida Molecular Pathology Core. Histopathology slides were digitally scanned by a BZ-X710 all-in-one fluorescence microscope (Keyence, Osaka, Japan). Inflammation and PAS-D globules were scored as the percentage of infiltratory area or PAS-D globule occupied area in total liver tissue area on the entire slide. Imaging quantification analysis was performed by BZ-X analysis software (Keyence, Osaka, Japan).

### Statistical analysis

Statistically significant differences were assessed by the student's t-test (parametric or nonparametric test) to compare two groups. One-way or two-way ANOVA was used to compare the means among three or more groups. Results are presented as mean ± SD. The criterion for statistical significance was set at $^*$ $p < 0.05$, $^{**}$ $p < 0.01$, $^{***}$ $p < 0.001$.

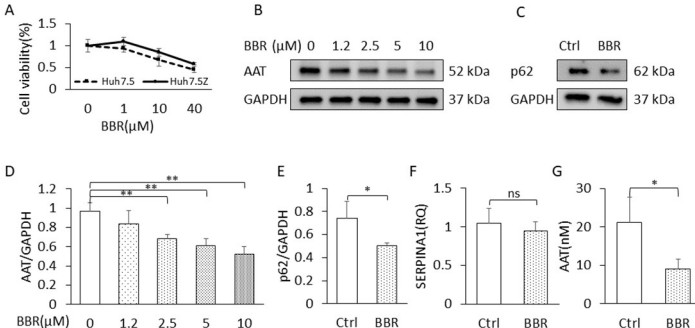

**Fig 1. BBR reduced intracellular AAT level in Huh7.5Z cells.** (A) MTT-based cell viability assay. Huh7.5 and Huh7.5Z cells were treated with 0 (served as control), 1, 10 and 40 μM of berberine chloride for 48 hours (n = 3). Cell viability (%) = (Mean OD sample / Mean OD Control) ×100. (B, D) Intracellular AAT load. Huh7.5Z cells were treated with 0, 1.2, 2.5, 5, and 10 μM of berberine chloride for 48 hours (n = 3), and western blot was performed to measure intracellular AAT. (B) Representative western blot and (D) quantification graph. (C, E, F, G) AAT transcription, degradation, and secretion. Huh7.5Z cells were treated with 2.5 μM of berberine chloride for 48 hours (n = 3), and western blot was performed for detection of p62. (C) Representative western blot and (E) quantification graph. (F) SERPINA1 gene expression was measured by qPCR, and (G) AAT concentration in culture medium was measured by ELISA. BBR: berberine. MTT: 3-(4,5-dimethylthiazol-2-yl)-2,5-diphenyl-2H-tetrazolium bromide. qPCR: quantitative polymerase chain reaction. ELISA: enzyme-linked immunosorbent assay. SERPINA1: Serine protease inhibitor clade A member 1 gene (encoding alpha-1 antitrypsin).

## Results

### BBR reduced cell viability in Huh7.5 and Huh7.5Z cells but no Pi*Z-specific toxicity was detected

Huh7 is a hepatocarcinoma cell line, and BBR exhibits antitumor activity. It has been reported that BBR reduces cell viability in Huh7 cells [33]. In addition, to investigate if BBR has Pi*Z-specific toxicity, we treated Huh7.5Z cells and Huh7.5 cells with 1, 10, and 40 μM of berberine chloride or the same volume of DMSO as a control for 48 hours. MTT-based cell viability was calculated as mean OD of sample divided by mean OD of untreated. The results were presented as mean ± SD (n = 3). Comparisons between BBR concentrations and between Huh7.5 and Huh7.5Z cells were performed by two-way ANOVA. The result of the statistical analysis showed that BBR reduced cell viabilities in both Huh7.5 and Huh7.5Z cells (p < 0.0001). Under BBR treatment, cell viabilities were higher in Huh7.5Z cells compared with Huh7.5 cells (p = 0.0063) (Fig 1A). Huh7.5 cells grow faster than Huh7.5Z cells [22]; therefore, they may be more sensitive to BBR treatment compared with Huh7.5Z cells. At least, BBR did not exhibit Pi*Z-specific cell toxicity in this experiment. At 10 μM of BBR concentration, the Huh7.5Z viability was 85%, which is acceptable for a pharmacological study. Therefore, we chose 1.2 to 10 μM as the range of dosage for the following in vitro studies.

### BBR reduced intracellular AAT levels in Huh7.5Z cells

To investigate the therapeutic potential of BBR in Pi*Z hepatocytes, we treated Huh7.5Z cells with 1.2, 2.5, 5, and 10 μM of BBR for 48 hours. Western blotting analysis showed that at the doses equal to or greater than 2.5 μM, BBR significantly reduced intracellular AAT protein loads, and the p values at doses of 1.2, 2.5, 5, and 10 μM were 0.238, 0.007, 0.006, and 0.003, respectively (Fig 1B and 1D, and images in S2A Fig in S1 File). Then we treated Huh7.5Z cells with 2.5 μM of BBR or the same volume of DMSO for 48 hours, SERPINA1 RT-qPCR result showed that there was no significant difference in the RNA levels between control and BBR groups (p = 0.513), indicating that BBR did not suppress SERPINA1 gene expression, at least

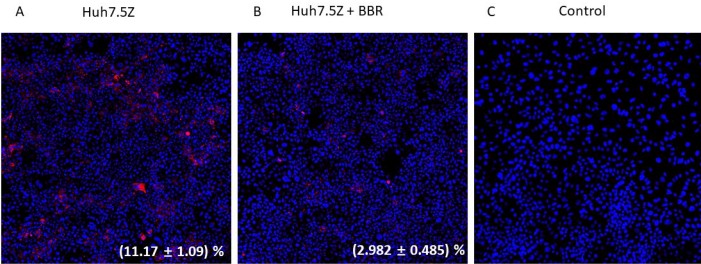

**Fig 2. BBR reduced AAT polymer in Huh7.5Z cells.** Huh7.5Z cells were treated with 2.5 μM of berberine chloride for 48 hours (n = 3), Immunofluorescence staining using 2C1 (red, an AAT polymer-specific antibody) showed decreased percentage of 2C1-positive cells in the BBR treatment group (B) compared with the untreated group (A). (C) Negative controls omitting the primary antibody.

at the transcriptional level (Fig 1F). ELISA result showed that the AAT levels in cell culture medium in BBR-treated wells were decreased (p = 0.024) (Fig 1G), indicating BBR did not enhance AAT secretion from Huh7.5Z cells. Significant decrease of p62 protein (p = 0.038) in BBR-treated cells was detected by western blot (Fig 1C and 1E, and images in S2B Fig in S1 File) indicating enhanced autophagy. Most likely, the reduction of intracellular AAT levels was due to increased degradation.

To evaluate the effect of BBR on intracellular AAT polymer, which contributes to the AAT aggregation in the ER, we treated Huh7.5Z cells with 2.5 μM of BBR for 48 hours, then we stained the AAT polymer with 2C1 antibody. The result showed that BBR treatment significantly reduced the percentage of 2C1-positive cells from (11.17 ± 1.09) % to (2.982 ± 0.485) %, p = 0.0002 (Fig 2).

## BBR exacerbated liver inflammation in PI*Z hAAT transgenic mice

We treated Pi*Z transgenic mice with BBR as aforementioned. As a pathological aspect, focal inflammatory infiltrations were observed in the liver of untreated Pi*Z mouse on the hematoxylin and eosin (H&E) staining slides (Fig 3A). Consistent with the distribution of PAS-D globules, most of the infiltratory foci were distributed in zone 2. Lesions were varied in size but predominantly were less than 100 μm. Increased and large (few hundreds μm) inflammatory foci were observed in the liver of BBR-treated Pi*Z mice. Most of the large inflammatory infiltratory lesions were appeared in pericentral area and extended to the lobules (Fig 3B and 3D). Considering the sample size, we combined the data from two experiments together to calculate the percentage of the inflammatory areas in the liver of BBR-treated and untreated Pi*Z mice. Significant increases in percentage of inflammatory areas in the liver were observed in BBR-treated mice in both female and male mice (p = 0.023 and 0.041, respectively) (Fig 3C, and date in S2 File), although only mild hAAT aggregation inflammation are presented in the liver of male mice in this strain of Pi*Z transgenic mice (Fig 7B, 7D).

The 2C1 immunohistochemistry stain in liver sections showed that the inflammatory infiltrations were initiated either surrounding AAT globules, otherwise known as PAS-D globules (Fig 4A), which mainly distribute in zone 2, or adjacent to central veins, i.e., zone 3 (Fig 4B and 4C). The 2C1-positive hepatocytes are mainly located in zone 3 [22]. The infiltrates surrounding globules were observed in both untreated and BBR-treated Pi*Z mice. Pericentral (zone 3) foci were predominantly observed in BBR-treated mice. Some infiltratory cells at pericentral area presented strong 2C1-positive signal in their cytoplasm (Fig 4D), implicating that the AAT polymers were engulfed by these cells.

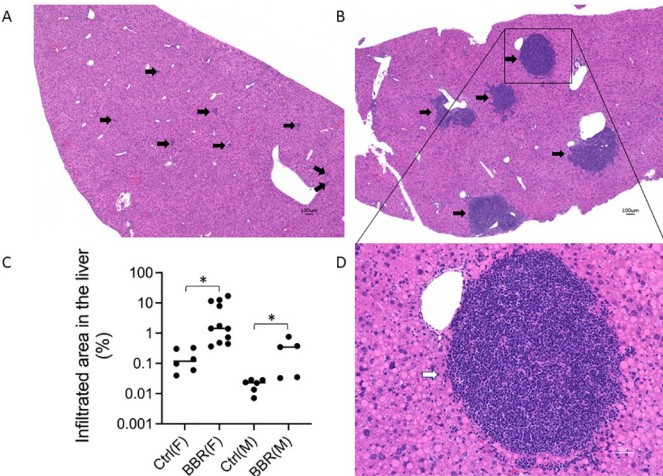

**Fig 3. BBR potentiated liver inflammation in the PI*Z hAAT transgenic mice.** The Pi*Z transgenic mice were treated with 50 mg/kg/day of berberine chloride (n = 7, 5 females, 2 males) or solvent only (n = 7, 4 females, 3 males) for 30 days. H&E staining was performed on the liver sections. (A) Focal inflammatory infiltrations (generally focus size was less than 100 μm) in the liver sections of untreated Pi*Z mice. (B) and (D) present larger inflammatory infiltratory lesions in the liver sections of BBR-treated Pi*Z mice. Most of the lesions are in the pericentral areas. (C) The quantification of the percentage of infiltrated area in the BBR-treated and the untreated Pi*Z mice from two *in vivo* experiments. The data from male and female mice were calculated separately because very rare inflammations were observed in male mice in both BBR-treated and untreated mice. Arrow: inflammation focus. Pi*Z: *protease inhibitor (SERPINA1) Z* allele. H&E: hematoxylin and eosin.

Inflammatory infiltrations in the liver of BBR-treated mice were composed of predominantly T and B lymphocytes. Immunohistochemistry staining showed that most of the infiltratory cells were CD3 (Fig 5C, and image in S4C Fig in S3 File) or B220 (Fig 5D, and image in S4D Fig in S3 File)-positive. Some macrophages (F4/80-positive) were seen in the peripheral

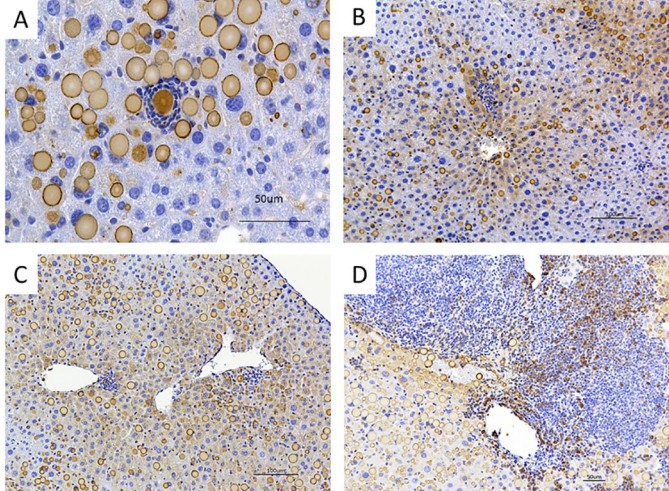

**Fig 4. Inflammation infiltrations were initiated surrounding AAT globules or pericentral area.** Representative images of immunohistochemical staining of 2C1 in the liver section of untreated and BBR-treated Pi*Z mice. (A) AAT globule-surrounding infiltrations were observed in the liver of both untreated and BBR-treated Pi*Z mice. (B, C) Inflammatory infiltrations initiated at pericentral areas (zone 3) in the liver sections of BBR-treated Pi*Z mice. These infiltrations are colocalized with 2C1-positive hepatocytes. (D) Inflammatory infiltration in the liver section from a BBR-treated Pi*Z mouse. Some infiltratory cells were 2C1-positive.

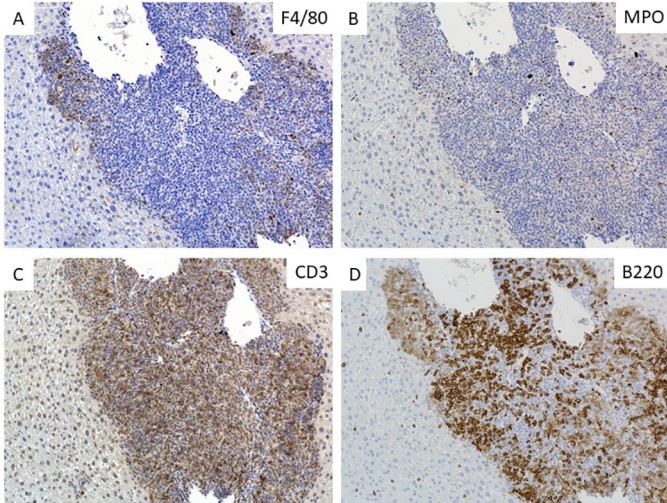

**Fig 5. The inflammatory infiltrate was composed of predominantly lymphocytes.** Pi*Z transgenic mice were treated with 50 mg/kg/day of berberine chloride (n = 7, 5 females, 2 males) for 30 days. Representative images of immunohistochemical staining of F4/80 (A), MPO (B), CD3 (C), and B220 (D) in the liver slides of BBR-treated Pi*Z mice.

areas of the lesions (Fig 5A, and image in S4A in S3 File). Neutrophils (MPO-positive) were not dominant in the lesions at the endpoint of the treatment (Fig 5B, and image in S3 File).

## BBR potentiated liver fibrosis in Pi*Z hAAT transgenic mice

Normally apparent fibrosis is not seen in the liver of untreated Pi*Z transgenic mice (Fig 6A) despite the significant human AAT aggregation in the liver [22]. Remarkable collagen depositions were detected with Picrosirius Red staining in liver sections of BBR-treated Pi*Z transgenic mice (Fig 6B and 6C). The fibrosis is not isolated around the portal area but presented in all liver zones. Fig 6D presented representative fibrosis in the liver sections of BBR-treated Pi*Z mice with H&E staining.

## BBR treatment did not reduce the PAS-D-positive globules in the liver of Pi*Z hAAT transgenic mice

Periodic acid-Schiff diastase (PAS-D)-positive cytoplasmic globules are a notable characteristic of AATD, which represent the retention of misfolded AAT. PAS-D globules can be used to evaluate the accumulated AAT burden in the liver [34]. Although in vitro study showed that BBR reduced intracellular AAT levels as aforementioned, compared with the untreated mice (Fig 7A and 7B), BBR treatment did not reduce PAS-D globules in Pi*Z transgenic mice (Fig 7C and 7D). The quantitative comparison between treated and untreated females was plotted as Fig 7E, p = 0.603. In this Pi*Z mouse model, at a similar blood hAAT concentration, male mice have much less PAS-D globules in their liver. Therefore, we did not include male mice data in the comparison of untreated and treated mice.

Interestingly, the percentage of the infiltrated area in the liver is correlated with the percentage of the PAS-D globules (Fig 7F) (r = 0.8929, p = 0.0123), indicating that the severity of inflammation depends on the severity of AAT aggregation.

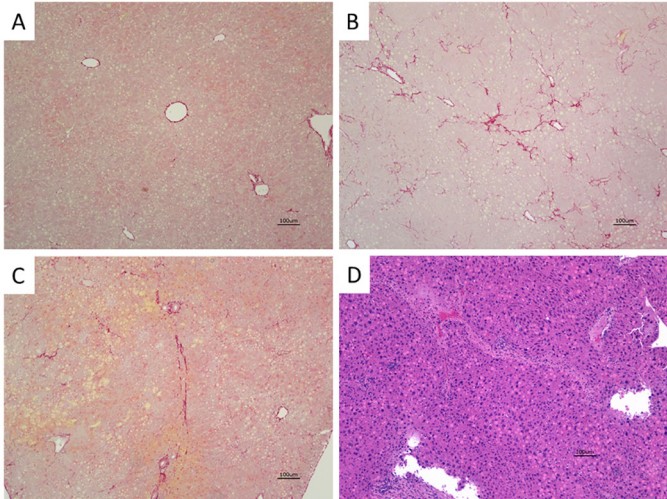

**Fig 6. BBR induced liver fibrosis in the Pi*Z transgenic mice.** Picrosirius Red staining was performed on the liver sections. Representative images present fibrosis in the liver sections of the BBR-treated Pi*Z mice (B, C) and a liver section of an untreated Pi*Z mouse as control (A). (D) Representative advanced fibrosis in the liver section of BBR-treated Pi*Z mice with H&E staining.

## BBR rebooted UPR and mTOR in the liver of Pi*Z hAAT transgenic mice

Previously we reported that in the liver of the Pi*Z transgenic mouse, the unfolded protein response (UPR) has been selectively suppressed [22], and we found the activity of the mammalian target of rapamycin (mTOR) complex 1 is also downregulated in another study (Yuanqing Lu et al.[Unpublished]). In the current study, we investigated these biological pathways by western blot in female mice since males only present very mild pathological changes in the

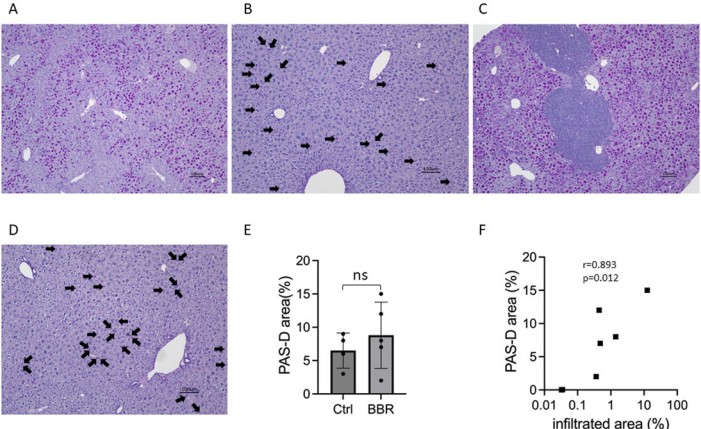

**Fig 7. BBR treatment did not reduce the PAS-D positive globules in the liver, and the severity of liver inflammation was correlated with the percentage of the PAS-D positive globules in the liver section.** Pi*Z transgenic mice were treated with 50 mg/kg/day of berberine chloride (n = 7, 5 females, 2 males) for 30 days. Representative images showed the PAS-D positive globules in the liver sections with PAS-D staining from (A) untreated females, (B) untreated males, (C) BBR-treated females, and (D) BBR-treated males. (E) Quantification of PAS-D globules between BBR-treated and untreated female Pi*Z mice. (F) The correlation of the percentage of infiltrated area in the liver section and the PAS-D globules area in the liver section. Arrow indicates PAS-D positive globules in males. PAS-D: periodic acid-Schiff diastase.

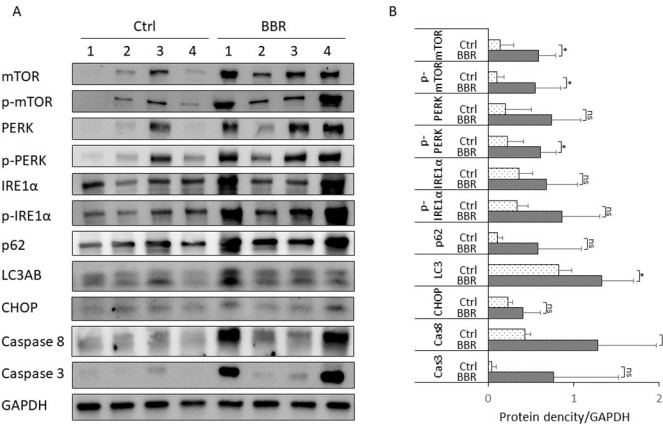

**Fig 8. The mTOR, UPR, and apoptosis systems were activated while autophagy degradation was impaired in some of the BBR-treated Pi*Z mice.** Pi*Z transgenic mice were treated with 50 mg/kg/day of berberine chloride (n = 7, 5 females, 2 males) for 30 days. Proteins were extracted from mouse liver tissues. Western blotting was achieved with protein from 4 untreated female mice and 4 BBR-treated female mice. (A) The representative blot images of mTOR, p-mTOR, PERK, p-PERK, IRE1α, p- IRE1α, p62, LC3AB, Caspase 8, and Caspase 3. (B) Quantification graph of western blot results. Western blot results were normalized by the GAPDH levels in the same blot and are presented as mean ± SD. Significances were determined by t-test. *p < 0.05. mTOR: the mammalian target of rapamycin; p-mTOR: phosphorylated mTOR; PERK: protein kinase R-like endoplasmic reticulum kinase; p-PERK: phosphorylated PERK; IRE1α: inositol-requiring enzyme 1; p- IRE1α: phosphorylated IRE1α; LC3AB: autophagy marker light chain 3 A and B.

liver (Fig 8A and 8B, and S4 File). All protein densities in this study were adjusted by the GAPDH loading controls on the same blot.

Our results showed that both mTOR and p-mTOR were significantly increased after BBR treatment. The mean ± SD of mTOR were 0.137 ± 0.160 in untreated mice and 0.592 ± 0.197 in BBR-treated mice, p = 0.012; the mean ± SD of p-mTOR were 0.100 ± 0.085 in untreated mice and 0.555 ± 0.295 in BBR-treated mice, p = 0.025.

We measured PERK and IRE1α branches of UPR, which were attenuated in the Pi*Z mouse liver according to our previous study [22]. Western blotting analysis showed p-PERK was increased in BBR-treated mice (mean ± SD = 0.611 ± 0.181) compared with untreated mice (mean ± SD = 0.224 ± 0.187), p = 0.025. Although there were no statistically significant increases in the means of PERK, IRE1α and p-IRE1α in BBR-treated mice, remarkable increases (> mean + 2xSD of the protein concentration in the untreated mice) in these proteins were observed in some BBR-treated individuals.

As an indicator of autophagy, the consumption of p62 was not observed in the BBR-treated mice compared with the untreated mice (p = 0.121). In some BBR-treated individuals the concentration of p62 was increased (> mean + 2xSD of the protein concentration in the untreated mice), indicating impaired autophagy, even though the LC3AB was increased in the BBR-treated mice (mean ± SD = 1.328 ± 0.376 while the mean ± SD in untreated mice = 0.827 ± 0.151, p = 0.048).

Caspase 8 was increased in the BBR-treated mice (mean ± SD = 1.286 ± 0.681 while in untreated mice the mean ± SD = 0.433 ± 0.062, p = 0.047). In 2 of 4 BBR-treated female mice, the levels of caspase 3 and CHOP were increased (> mean + 2xSD of the protein concentration in the untreated mice). However, due to the high level of variation in the treatment group, there were no significant differences in caspase 3 and CHOP between control and treatment groups when comparing the mean ± SD by student's t-test.

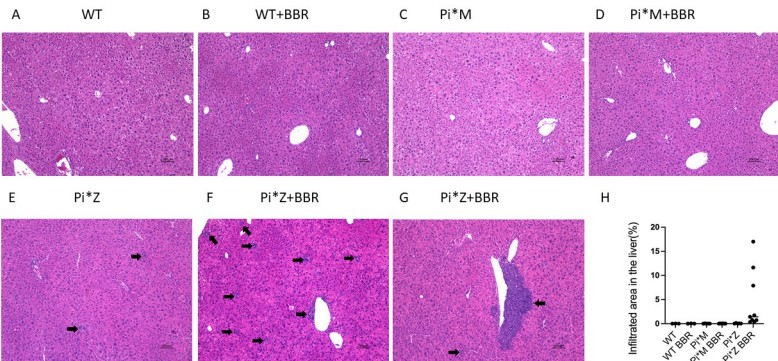

**Fig 9. BBR potentiated liver inflammation only in Pi\*Z mice.** WT, Pi\*M and Pi\*Z transgenic mice were treated with 50 mg/kg/day of BBR or solvent only for 30 days. The sample size was as follows: WT control n = 3, WT BBR n = 3, Pi\*M control n = 5, Pi\*M BBR n = 6, Pi\*Z control n = 5, and Pi\*Z BBR n = 9. H&E staining was performed on the liver section. Representative images of (A) WT control, (B) WT BBR, (C) Pi\*M control, (D) Pi\*M BBR, (E) Pi\*Z control, (F, G) Pi\*Z BBR. (H) Quantification of the percentages of infiltrated area in the liver. Arrow indicates inflammation focus.

## The effect of exacerbation of liver inflammation by BBR treatment was Pi\*Z-specific

To confirm the results of the above animal experiment and examine if the effect of provoking liver inflammation by BBR was Pi\*Z-specific, we repeated the above animal experiment in Pi\*Z AAT transgenic mice. Meanwhile, we included WT and Pi\*M AAT transgenic mice in this experiment as well. The results indicated that BBR only induced increased inflammation in the Pi\*Z mice, but not in Pi\*M and WT mice (Fig 9). The large (> 100 μm) and mainly peri-central infiltratory lesions were only seen in the BBR-treated Pi\*Z mice. The percentage of infiltration area in the liver sections was 0.35% to 17.01% in the BBR-treated Pi\*Z mice, while it was 0.02% to 0.13% in the untreated Pi\*Z mice. This percentages in both BBR-treated and untreated Pi\*M mice were the same (0.01%–0.06%), as similarly, it was 0.01%–0.04% in untreated WT mice and 0.01%–0.06% in BBR-treated WT mice. Brown-Forsyth test (Unequal SD ANOVA) found a significant difference between BBR-treated and untreated Pi\*Z mice, p = 0.021, as shown in Fig 9H.

## Discussion

The environmental risk factors of AATD-associated liver disease remain unclear. Discovery of substances, especially these apparently harmless dietary supplements that have potential liver toxicity for AATD patients, is important to reduce the prevalence of liver disease in adult AATD. So far, nonsteroidal anti-inflammatory drugs (NSAIDs) are the only class of medications that has been reported to exacerbate liver damage in an AATD murine model [35]. In the present study, we discovered that BBR, a widely available over-the-counter dietary supplement, may be another risk factor to trigger AATD-associated liver damage.

Certain levels of liver inflammation can be detected in Pi\*Z mice. Our observation suggests that the infiltrated immune cells were attracted by the PAS-D-positive globules. Liver inflammation can be significantly exacerbated by BBR treatment in Pi\*Z mice. This BBR liver toxicity appears to be Pi\*Z disease-specific, since it was not observed in WT and Pi\*M mice. According to the literature, the dose we used in this study is safe for WT mice [36]. Liver damage has not been observed in previous study even with 300 mg/kg/day of BBR oral administration for 14 days in C57BL/6 mice [37], which is the genetic background of our Pi\*Z and Pi\*M mice.

BBR-potentiated liver inflammation seems to be associated with ER stress. First, BBR-induced inflammation predominantly presented in zone 3, and in this zone, the hepatocytes are AAT polymer stain (2C1)-positive. Thus, these hepatocytes should suffer more ER stress. Second, the UPR signaling pathways were activated in BBR-treated Pi*Z mice, which is in line with a previous study using a cellular model [38]. The detailed mechanisms underlying how BBR provokes UPR remains unknown. UPR plays a prominent role in the pathogenesis of chronic liver disease by triggering the innate immune response and inducing cell death [39–42]. UPR also activates mTOR [43], which may enhance AAT accumulation by inhibiting autophagic degradation [44] and enhancing protein synthesis.

A question is raised in this study: why do Huh7.5Z cells and Pi*Z mice respond to BBR treatment differently? We assume that the different responses to BBR treatment were due to the different UPR status between these two models. Huh7.5 cells exhibit a high proliferation rate (G0% = 19%) [45]. There are many young Huh7.5Z cells (about 80%) in which the accumulated AAT polymers have not reached the detectable level [22]. Under this mild ER stress, we suspect that the UPR enhanced by BBR will mainly be a pro-survival event. However, with a long lifespan, about 200 to 400 days in adult ICR mice [46], most of the hepatocytes in Pi*Z transgenic mice are under prolonged ER stress [22]. Suppressed UPR and prolonged AAT accumulation in the hepatocytes leads to severe ER stress. It is plausible to speculate that with advanced AAT accumulation, the UPR rebooted by BBR in the hepatocytes of Pi*Z mice will most likely exhibit a pro-apoptotic property.

Previously, we reported that in Pi*Z mice, the attenuated UPR can be rebooted by LPS treatment, and then followed by a recovery to its attenuated levels after a couple of days [22]. These UPR phases may not be in sync between individuals in a cohort of animals under prolonged treatment. In addition, a mathematical UPR model showed that in certain ER stress status, the PERK downstream proteins, such as ATF4 and CHOP, exhibit oscillations [47]. These behaviors perhaps explained the large variations of the levels of UPR and its target proteins in BBR-treated Pi*Z mice at the endpoint of the treatment.

Liver fibrosis is a major characteristic of AATD-associated liver disease. It has not been observed in this Pi*Z transgenic murine model, even with one dose of LPS treatment [22]. In this study, apparent liver fibrosis was observed in BBR-treated Pi*Z mice, which further confirmed that BBR can potentiate typical AATD-associated liver injury. This animal experiment may serve as an AATD-associated liver disease model.

In conclusion, BBR ameliorated Pi*Z AAT aggregation in Huh7.5Z cells but potentiated liver inflammation and induced fibrosis in the liver of Pi*Z transgenic mice. These adverse effects were only observed in Pi*Z mice. Avoidance of BBR for AATD patients is advised. BBR liver toxicity in Pi*Z mice are thought to be related to the activation of UPR and mTOR pathways. Further studies are needed to fully understand the underlying molecular mechanisms, which will be beneficial to discover more substances that may cause liver damage in AATD patients.

## Supporting information

**S1 Table. List of antibodies used for western blotting analysis.**
(DOCX)

**S1 File. Supporting information for Fig 1B and 1C.**
(DOCX)

**S2 File. Supporting information for Fig 3C.**
(DOCX)

**S3 File. Supporting information for Fig 5.**
(DOCX)

**S4 File. Supporting information for Fig 8.**
(DOCX)

**S5 File. Lu BBR Raw data.**
(XLSX)

## Author Contributions

**Conceptualization:** Yuanqing Lu, Mark L. Brantly.

**Data curation:** Yuanqing Lu.

**Formal analysis:** Yuanqing Lu.

**Funding acquisition:** Mark L. Brantly.

**Investigation:** Yuanqing Lu, Naweed S. Mohammad.

**Methodology:** Yuanqing Lu.

**Resources:** Alek M. Aranyos, Mark L. Brantly.

**Software:** Yuanqing Lu.

**Supervision:** Mark L. Brantly.

**Validation:** Yuanqing Lu.

**Visualization:** Yuanqing Lu.

**Writing – original draft:** Yuanqing Lu.

**Writing – review & editing:** Yuanqing Lu, Naweed S. Mohammad, Jungnam Lee, Karina A. Serban, Mark L. Brantly.

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
