## [Decision Letter · Decision Letter 0]

16 May 2024

PONE-D-24-12616Berberine potentiates liver inflammation and fibrosis in the PI*Z hAAT transgenic murine modelPLOS ONE

Dear Dr. Brantly,

Thank you for submitting your manuscript to PLOS ONE. After careful consideration, we feel that it has merit but does not fully meet PLOS ONE’s publication criteria as it currently stands. Therefore, we invite you to submit a revised version of the manuscript that addresses the points raised during the review process.

This is an interesting and important study question, along with a great point on the discordance of *in vitro* vs. *in vivo* models. You will see that the reviewers have several major points that need to be addressed; some of these can be addressed with an explicit limitations section. If you have banked samples (*e.g.*, for liver chemistry) it would be ideal to include these analyses given the nature of the topic (safety and liver toxicity). Because of what appears might be a sex-specific response (which is also not surprising for this topic), the issue with sample size in the animal experiments is a substantial concern. You should know at this stage, that while I think this work is important, this specific point makes me uncomfortable endorsing the manuscript further if sample size cannot be addressed. Other points from me that should be examined are as follows: Are the images in figure 5 consecutive sections from the same liver? Is that truly representative if so?There should be more said about the chosen *in vivo* dose either in the methods, discussion, or both. This should bein the context of other studies, what people typically would consume as a supplement, and also to the fact that BBR does not have good oral bioavailability. REF 33 somewhat gets at this as you already mentioned, and I understand that oral is the appropriate route (since that is how people consume it), but these are all factors that are important to consider for translational context.The biochemical/mechanistic literature on glucose metabolism is probably the richest set of detailed cell biology information available for BBR. I was surprised not to see this synthesized in the context of a liver-specific study, particularly where a major sensor of cellular metabolism is discussed.Please submit your revised manuscript by Jun 30 2024 11:59PM. If you will need more time than this to complete your revisions, please reply to this message or contact the journal office at plosone@plos.org. Please include the following items when submitting your revised manuscript:A rebuttal letter that responds to each point raised by the academic editor and reviewer(s). You should upload this letter as a separate file labeled 'Response to Reviewers'.A marked-up copy of your manuscript that highlights changes made to the original version. You should upload this as a separate file labeled 'Revised Manuscript with Track Changes'.An unmarked version of your revised paper without tracked changes. You should upload this as a separate file labeled 'Manuscript'.

We look forward to receiving your revised manuscript.

Sincerely,

Nicholas A. Pullen, Ph.D.

Academic Editor

PLOS ONE

Journal Requirements:

"MB. The Alpha-1 Foundation Research Professorship #007319."

5. We note that you have referenced (unpublished manuscript) on page 25,  which has currently not yet been accepted for publication. Please remove this from your References and amend this to state in the body of your manuscript: (ie “Bewick et al. [Unpublished]”) as detailed online in our guide for authors

Additional Editor Comments:

This is an interesting and important study question, along with a great point on the discordance of in vitro vs. in vivo models. You will see that the reviewers have several major points that need to be addressed; some of these can be addressed with an explicit limitations section. If you have banked samples (e.g., for liver chemistry) it would be ideal to include these analyses given the nature of the topic (safety and liver toxicity). Because of what appears might be a sex-specific response (which is also not surprising for this topic), the issue with sample size in the animal experiments is a substantial concern. You should know at this stage, that while I think this work is important, this specific point makes me uncomfortable endorsing the manuscript further if sample size cannot be addressed. Other points from me that should be examined are as follows:

- Are the images in figure 5 consecutive sections from the same liver? Is that truly representative if so?

- There should be more said about the chosen in vivo dose either in the methods, discussion, or both. This should bein the context of other studies, what people typically would consume as a supplement, and also the fact that BBR does not have good oral bioavailability. REF 33 somewhat gets at this as you already mentioned, and I understand that oral is the appropriate route (since that is how people consume it), but these are all factors that are important to consider for translational context.

- The biochemical/mechanistic literature on BBR effects on glucose metabolism is probably the richest set of information available. I was surprised not to see this synthesized in the context of a liver-specific study.

Reviewers' comments:

Reviewer's Responses to Questions

**Comments to the Author**

1. Is the manuscript technically sound, and do the data support the conclusions?

Reviewer #1: Yes

Reviewer #2: Yes

2. Has the statistical analysis been performed appropriately and rigorously? 

Reviewer #1: Yes

Reviewer #2: Yes

3. Have the authors made all data underlying the findings in their manuscript fully available?

Reviewer #1: Yes

Reviewer #2: Yes

4. Is the manuscript presented in an intelligible fashion and written in standard English?

Reviewer #1: Yes

Reviewer #2: Yes

5. Review Comments to the Author

Reviewer #1: Lu et al. investigated the therapeutic effects and toxicities of berberine, a commonly used nature dietary supplement, on alpha-1 antitrypsin deficiency, using transgenic Pi*Z hepatocytes and Pi*Z mice. It a very interesting approach since there is evidence that berberine has both negative and positive effects on the liver. In my point of view, the study adds new and important information to the role of berberine on diseased individuals. There are some points to be clarified:

1) A major limitation is the low number of animals used in the study (n=7 including males and females) which is aggravated by the fact that there is a high level of variation in the BBR treated group and also that the observed effects appear to be gender dependent (see fig 3). This issue could be resolved by increasing the number of animals in each group.

2) It is well known that altered mitochondrial physiology and decreased ATP availability has been associated with some properties of berberine (antitumoral, antihyperglycemic and maybe toxic effects). Do authors think that the decrease in ATP content could also be related with the toxic effects observed in this study?

3) Fig 2 – resolution was low, difficult to see the red points.

4) Fig 7 and 9 – The white arrows used are confusing with the histological image. Perhaps using black arrows would be better.

Reviewer #2: Alpha1-antitrypsin deficiency (AATD) is an inherited disorder arising due to mutations in alpha1-antitrypsin (AAT) gene leading to its polymerization and consecutive proteotoxic liver injury. Since there are no approved treatments for AATD-associated liver disease, Lu et al. tested the potential usefulness of berberine, a compound widely used in alternative medicine. This is a nice piece of work, however, a couple of issues need to be addressed prior to publication. In particular:

- The authors used RIPA buffer to measure intracellular AAT levels. Given that polymerized AAT is highly insoluble, it would be useful to check, whether RIPA buffer solubilizes as much AAT as buffers with higher SDS amounts. If not, this fact should be mentioned as a limitation.

- Line 303-the statement that reduction of intracellular AAT levels was due to enhancing of the autophagy is not supported by data and should be tuned down. For example, one could say that it might be due to increased degradation.

- It would be useful to report serum liver enzyme levels as a surrogate of hepatocellular injury.

- It would be useful to use a second method of fibrosis quantification/stellate cell activation (i.e. collagen RT-PCR, hydroxyproline assay or a comparable technique).

- In addition to PAS-D staining, it would be nice to quantify the intrahepatic AAT levels in treated/non-treated animals via immunoblotting.

6. PLOS authors have the option to publish the peer review history of their article (what does this mean?). If published, this will include your full peer review and any attached files.

Reviewer #1: No

Reviewer #2: No

---

## [Author Response · Author response to Decision Letter 0]

31 Jul 2024

PONE-D-24-12616

Berberine potentiates liver inflammation and fibrosis in the PI*Z hAAT transgenic murine model

PLOS ONE

Dear Professor Pullen, 

Below is a point by point response to the reviewers comments. We appreciate the opportunity to resubmit our manuscript and believe the reviewers’ and the Editor’s suggestions have allowed us to improve the manuscript. Reponses are in BOLD.

• Are the images in figure 5 consecutive sections from the same liver? Is that truly representative if so?

Yes. In Figure 5 the sections are from the same liver, and they are representative. To support this, we present the same immunohistochemical staining images as Figure 5 but from another BBR-treated Pi*Z mouse on S4. We believe that the macrophages play a critical role in initiation of inflammation and are then replaced by lymphocytes when we harvest livers. 

• There should be more said about the chosen in vivo dose either in the methods, discussion, or both. This should be in the context of other studies, what people typically would consume as a supplement, and also to the fact that BBR does not have good oral bioavailability. REF 33 somewhat gets at this as you already mentioned, and I understand that oral is the appropriate route (since that is how people consume it), but these are all factors that are important to consider for translational context.

We totally agree with you. We updated the method and discussion sections accordingly. 

• The biochemical/mechanistic literature on glucose metabolism is probably the richest set of detailed cell biology information available for BBR. I was surprised not to see this synthesized in the context of a liver-specific study, particularly where a major sensor of cellular metabolism is discussed.

Yes. The beneficial effects of berberine on blood glucose and lipid profile have been confirmed by many in vivo studies in diabetic and metabolic syndrome animal models. We measured blood glucose levels at the end of the experiments, and there were no abnormal blood glucose levels observed. There was no difference in blood glucose levels between the BBR-treated and the control groups, at least at this age of the Pi*Z mice. The oil-red O scores in the liver of BBR-treated Pi*Z mice were slightly low compared with the untreated group but statistically there was no significant difference between the two groups.

Yes.

2. To comply with PLOS ONE submissions requirements, in your Methods section, please provide additional information regarding the experiments involving animals and ensure you have included details on (1) methods

---

## [Decision Letter · Decision Letter 1]

3 Sep 2024

Berberine potentiates liver inflammation and fibrosis in the PI*Z hAAT transgenic murine model

PONE-D-24-12616R1

Dear Dr. Brantly,

We’re pleased to inform you that your manuscript has been judged scientifically suitable for publication and will be formally accepted for publication once it meets all outstanding technical requirements.

Sincerely,

Nicholas A. Pullen, Ph.D.

Academic Editor

PLOS ONE

Additional Editor Comments (optional):

I have confirmed adequate effort has been made to address previous comments. Therefore, I am happy to recommend the manuscript for publication.

Reviewers' comments:

Reviewer's Responses to Questions

**Comments to the Author**

1. If the authors have adequately addressed your comments raised in a previous round of review and you feel that this manuscript is now acceptable for publication, you may indicate that here to bypass the “Comments to the Author” section, enter your conflict of interest statement in the “Confidential to Editor” section, and submit your "Accept" recommendation.

Reviewer #1: All comments have been addressed

Reviewer #2: (No Response)

2. Is the manuscript technically sound, and do the data support the conclusions?

Reviewer #1: Yes

Reviewer #2: Yes

3. Has the statistical analysis been performed appropriately and rigorously? 

Reviewer #1: Yes

Reviewer #2: Yes

4. Have the authors made all data underlying the findings in their manuscript fully available?

Reviewer #1: Yes

Reviewer #2: Yes

5. Is the manuscript presented in an intelligible fashion and written in standard English?

Reviewer #1: Yes

Reviewer #2: Yes

6. Review Comments to the Author

Reviewer #1: (No Response)

Reviewer #2: Some comments could not be answered due to insufficient manpower, but in my opinion, the manuscript is still useful for the audience of PLoS One. The author should use buffers with higher SDS content in their future experiments (such as Laemmli 4x reducing buffer).

7. PLOS authors have the option to publish the peer review history of their article (what does this mean?). If published, this will include your full peer review and any attached files.

Reviewer #1: No

Reviewer #2: No

---

## [Editor Report · Acceptance letter]

10 Sep 2024

PONE-D-24-12616R1 

PLOS ONE

Dear Dr. Brantly, 

I'm pleased to inform you that your manuscript has been deemed suitable for publication in PLOS ONE. Congratulations! Your manuscript is now being handed over to our production team.

Kind regards, 

on behalf of

Dr. Nicholas A. Pullen 

Academic Editor

PLOS ONE